DOI: 10.1038/s41467-018-04550-9　　OPEN

# Analysis of 3800-year-old *Yersinia pestis* genomes suggests Bronze Age origin for bubonic plague

Maria A. Spyrou [1,2], Rezeda I. Tukhbatova[1,3], Chuan-Chao Wang [1,4], Aida Andrades Valtueña[1], Aditya K. Lankapalli[1], Vitaly V. Kondrashin[5], Victor A. Tsybin[6], Aleksandr Khokhlov[7], Denise Kühnert[1,8], Alexander Herbig [1], Kirsten I. Bos[1] & Johannes Krause [1,2]

The origin of *Yersinia pestis* and the early stages of its evolution are fundamental subjects of investigation given its high virulence and mortality that resulted from past pandemics. Although the earliest evidence of *Y. pestis* infections in humans has been identified in Late Neolithic/Bronze Age Eurasia (LNBA 5000–3500y BP), these strains lack key genetic components required for flea adaptation, thus making their mode of transmission and disease presentation in humans unclear. Here, we reconstruct ancient *Y. pestis* genomes from individuals associated with the Late Bronze Age period (~3800 BP) in the Samara region of modern-day Russia. We show clear distinctions between our new strains and the LNBA lineage, and suggest that the full ability for flea-mediated transmission causing bubonic plague evolved more than 1000 years earlier than previously suggested. Finally, we propose that several *Y. pestis* lineages were established during the Bronze Age, some of which persist to the present day.

[1] Max Planck Institute for the Science of Human History, Kahlaische Str. 10, 07745 Jena, Germany. [2] Institute for Archaeological Sciences, University of Tübingen, Rümelinstrasse 23, 72070 Tübingen, Germany. [3] Center of Excellence "Archaeometry", Kazan Federal University, Kazan 420008, Russian Federation. [4] Department of Anthropology and Ethnology, Xiamen University, 361005 Xiamen, China. [5] LLC "Gefest", Michurina Str. 4, Samara 443030, Russia. [6] State Institute of Culture, Agency for Preservation of the Historical and Cultural Heritage of the Samara Region, Samara 443010, Russia. [7] Samara State University of Social Sciences and Education, Maxim Gorky Str., Samara 443090, Russia. [8] Department of Infectious Diseases and Hospital Epidemiology, University Hospital Zurich, 8091 Zurich, Switzerland. Correspondence and requests for materials should be addressed to M.A.S. (email: spyrou@shh.mpg.de) or to K.I.B. (email: bos@shh.mpg.de) or to J.K. (email: krause@shh.mpg.de)

Yersinia pestis, the causative agent of bubonic, pneumonic and septicaemic plague, evolved from the closely related environmental progenitor Y. pseudotuberculosis[1]. Although primarily a coloniser of sylvatic rodents via flea-dependent transmission, ancient DNA studies have demonstrated its status as an infectious disease in humans for the last 5000 years[2,3], and have confirmed its involvement in some of the most devastating historical pandemics[4,5]. The first historically recorded plague pandemic began with the Plague of Justinian (AD 541–543), and persisted until the eighth century AD[5,6]. The second pandemic occurred between the 14th and 18th centuries AD, began with the infamous Black Death of Europe in 1347[4,7] and was a precursor of modern-day plague epidemics over a wide geographic range[8].

Today, plague has a near-worldwide distribution and is maintained within sylvatic rodent populations[9]. Although several of these rodent reservoirs were established during the third plague pandemic that began in 19th century China[10,11], many of those identified in Central and East Asia and most notably those of the Caspian Sea region harbour Y. pestis strains that occupy basal positions in the global phylogeny (i.e., 0.PE2)[12]. This supports the idea of these foci having persisted for millennia[12–15]. What remains unknown, however, is the time period and processes involved in their establishment, and the level of Y. pestis genetic diversity harboured within them during the early phases of its evolution.

After its divergence from Y. pseudotuberculosis, Y. pestis acquired its high pathogenicity and distinct niche mainly by chromosomal gene loss[16] as well as the acquisition of two virulence-associated plasmids, pMT1 and pPCP1[1,17,18]. Throughout this process, one of the most crucial evolutionary adaptations related to its pathogenicity was its ability to colonise arthropods, a phenotypic/functional gain mediated by a combination of chromosomal and plasmid loci[19,20]. These genetic changes are central to the most common "bubonic" form of the disease, where bacteria enter the body via the bite of an infected flea, travel via the lymph to the closest lymph node and replicate while evading host defences. Recent ancient genomic investigations of Y. pestis have identified its earliest known variants in Eurasia during the Late Neolithic/Bronze Age period (LNBA) that show genetic characteristics incompatible with arthropod adaptation. These strains, therefore, have been considered incapable of an efficient flea-based transmission[2]; however, the alternative early-phase transmission could have provided an independent means of arthropod dissemination[2,3,21]. To date, the earliest evidence of a Y. pestis strain with signatures associated with flea adaptation has been reported during the Iron Age through shotgun sequencing of an ~2900-year-old genome from Armenia (strain RISE397), though at a coverage too low (0.25-fold) to permit confident phylogenetic positioning[2]. Although the mechanism by which the LNBA lineage caused human disease is unclear, its frequency in Eurasia during the Bronze Age[2,3] and its phylogeographic pattern that mimics contemporaneous human migrations are noteworthy[3].

The Bronze Age in Eurasia was a period of technological transition among human populations, often associated with the initiation of cultural and societal complexity[22]. Recent aDNA analysis of human remains from the time period between 5500 and 3200 BP has linked such transitions to a large-scale expansion of "Yamnaya" pastoralists from the Pontic–Caspian steppe both westwards into Europe, giving rise to the so-called "Corded–ware complex", and eastwards into Central Asia and the Altai region, represented by Early Bronze Age (EBA) cultures such as the "Afanasievo"[23,24]. Specifically in Europe, the "Yamnaya" migrations resulted in admixture with the local Neolithic farmer populations, forming the gene pool that appears to constitute European populations to the present day[23–25]. In addition, recent studies propose subsequent population expansions from Europe back into Central Asia during the Middle and Late Bronze Age (MLBA), which is genetically reflected by the appearance of European farmer-related ancestry among Late Bronze Age (LBA) steppe populations such as "Sintashta", "Srubnaya", "Potapovka" and "Andronovo"[24,26].

The central steppe region seems to have played a significant role as a migration corridor during the entire Bronze Age, and as such, it likely facilitated the spread of human-associated pathogens, such as Y. pestis, across Eurasia. Here, we explore additional Y. pestis diversity in that region by isolating strains from LBA Samara, in Russia. We identify a Y. pestis lineage contemporaneous to the LNBA strains with genomic variants consistent with flea adaptation. This reveals the co-circulation of two Y. pestis lineages during the Bronze Age with different properties in terms of their transmission and disease potentials.

## Results

**Y. pestis and human endogenous DNA screening**. We screened a total of 64 million shotgun next-generation sequencing (NGS) reads (Supplementary Table 1) from nine teeth of nine individuals recovered from kurgan burials in the Samara region (see Supplementary Methods) to assess the endogenous human DNA content and the possible presence of Y. pestis. Our Y. pestis screening procedure involved (1) mapping of all reads against the CO92 reference genome (NC_003143.1), and (2) taxonomically classifying the reads to bacterial species using the metagenomic tool MALT[27] with a special focus on those assigned to both the entire Y. pseudotuberculosis complex, as well as Y. pestis specifically (Supplementary Table 1). As MALT has previously proven to be an efficient tool in binning reads from complex metagenomic datasets into their respective bacterial taxa and has been successfully used for identifying pathogen DNA in archaeological material[27], we considered individuals as putatively positive only when reads were assigned to Y. pestis by both conventional read-mapping and MALT. Our screening revealed four potentially positive individuals (Supplementary Table 1), one of which, individual RT5 (Fig. 1a), exhibited the highest amounts of endogenous Y. pestis DNA (0.11%, Supplementary Table 1). Notably, individual RT5 also exhibited exceptional human DNA preservation (31.3% of endogenous DNA, Supplementary Table 2). A shotgun-sequencing approach was, therefore, used for retrieving the entire Y. pestis and human genomes from this specimen. In addition, an in-solution Y. pestis enrichment approach was employed for putatively positive individuals.

**Human uniparental and genomic analyses**. Shotgun sequencing of RT5 resulted in 1.14 billion raw reads and a 4.2-fold average human genomic coverage (Supplementary Table 3). Genetic sex identification assigned RT5 to a male, which is in line with the anthropological assignment (Fig. 1a, Supplementary Methods). Nuclear contamination estimates based on X-chromosomal heterozygosity were low in RT5, estimated at an average of 0.5% (Supplementary Table 4), which permitted the usage of all generated human data. Y-chromosomal and mitochondrial assignment revealed the individual carrying R1a1a1b and U2e2a haplogroups, respectively (Supplementary Methods, and Supplementary Table 5). To gain insight into the ancestry of RT5, we performed principal component analysis (PCA)[28–30] and ADMIXTURE[31] analysis, where previously published ancient Eurasian populations[32–35] were used as comparative datasets (Fig. 1 b, c). Our PCA (Fig. 1b) and ADMIXTURE (Fig. 1b, c) plots show RT5 to have close genetic affinity to ancient populations from EBA Europe and the MLBA steppe, which are

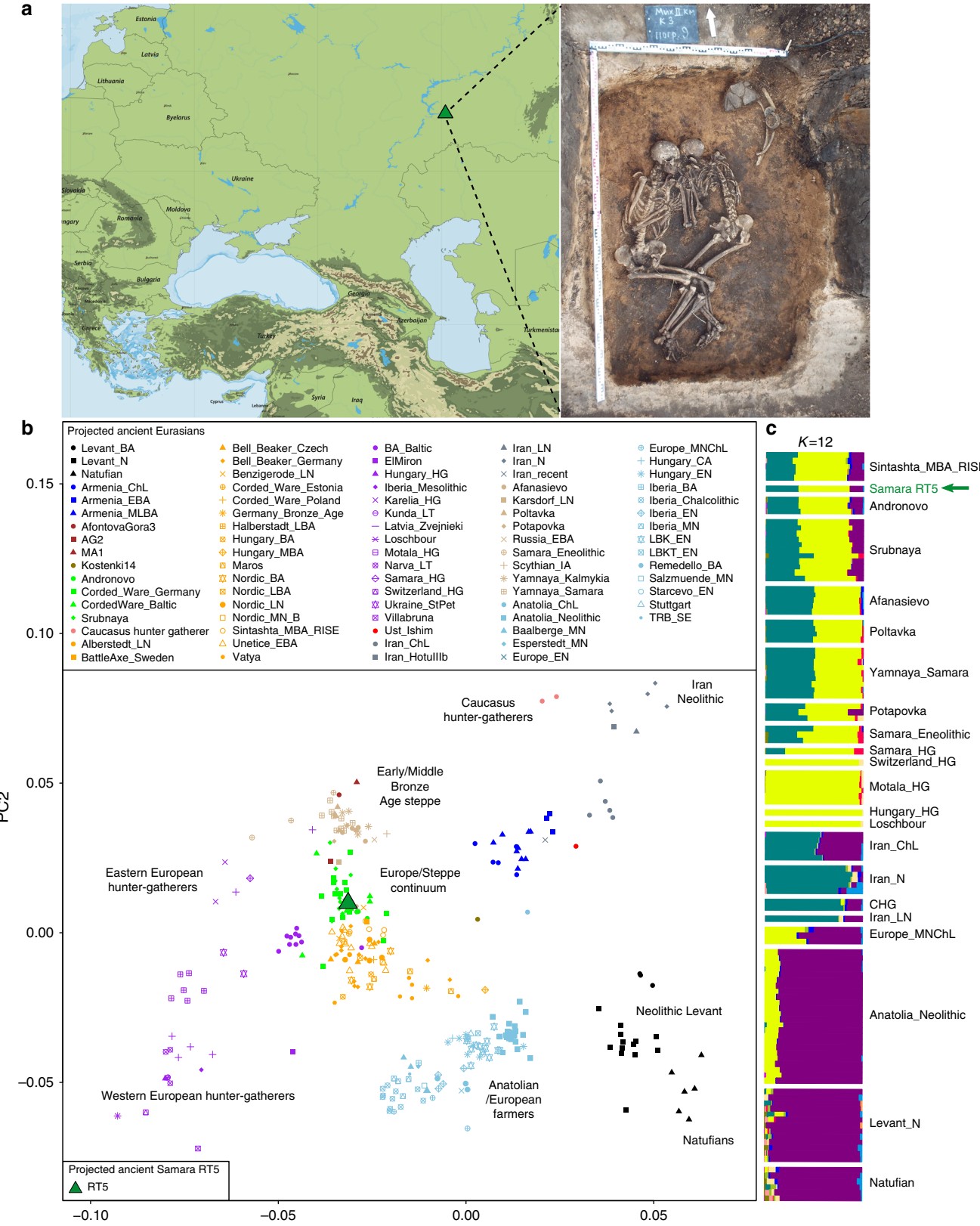

**Fig. 1** Population genetic analysis to infer the ancestry of RT5. **a** Geographic location (map purchased from vectormaps.de) and picture of RT5 burial in the Mikhailovsky II site (picture credits to V.V. Kondrashin and V.A. Tsybin). **b** Principal component analysis (PCA) of modern-day western Eurasian populations (not shown) and projected ancient populations ($n = 82$, see population labels), including the newly sequenced RT5 individual from Samara and **c** estimation of ancestral admixture components using ADMIXTURE analysis ($K = 12$) (see Supplementary Methods)

genetically distinct from EBA populations from the Central Asian steppe, as they encompass early European farmer-related ancestry as part of their genetic composition (Fig. 1c). Examples of such groups include the European "Corded Ware"-associated populations, the "Andronovo" from the Altai region and the Samara-region "Srubnaya" culture with which RT5 has been archaeologically associated.

**Y. pestis quality assessment and genome reconstruction.** Although *Y. pestis* has been previously identified in Bronze Age individuals[2,3], its presence in the Volga region "Srubnaya"-associated populations has not been characterised to date. After *Y. pestis* capture, samples RT5 and RT6 yielded an average genomic coverage greater than 1-fold, with RT6 reaching 1.9-fold and RT5 reaching 32.3-fold (Supplementary Table 3). In addition, we retrieved a 9.2-fold *Y. pestis* genome from RT5 shotgun sequencing alone (Supplementary Table 3). Comparison between the RT5-captured and deep-shotgun-sequenced *Y. pestis* reads revealed nearly identical GC contents and deamination profiles (Supplementary Table 3 and Supplementary Fig. 1), but significantly different read-length distributions (Supplementary Fig. 2), with an average fragment length increase of 3.6 bp after capture (*t*-test, *P*-value < 2.2e-16). Assessment of SNP profiles inferred from captured and shotgun-sequenced RT5 reads shows no SNP-calling inconsistencies between the two datasets (Supplementary Data 1).

Moreover, during a first assessment of coverage across the plasmids (Supplementary Table 3), it became apparent that RT5 and RT6 harbour the 1.8 kb Yersinia murine toxin (*ymt*) gene locus on the pMT1 plasmid (Fig. 2a), which encodes for a virulence factor essential for the colonisation of the flea's midgut. This gene is absent in all previously sequenced LNBA strains[2,3], though it has been identified in a later Iron Age individual (~2900 BP) from modern-day Armenia (RISE397) (Fig. 2a)[2].

**Y. pestis phylogenetic analysis.** In order to reconstruct a phylogeny using a maximum likelihood (ML) approach, we analysed our RT5 *Y. pestis* isolate against a total of 177 genomes, including previously published ancient strains, as well as a worldwide modern *Y. pestis* dataset (see Methods and Supplementary Data 2)[2–4,6,8,12,14,15,36]. *Y. pseudotuberculosis* (strain IP32953)[18] was used as an outgroup for rooting the tree. Our ML tree (Fig. 2b) is consistent with previously published phylogenies, where the LNBA isolates occupy the most basal *Y. pestis* branch[2,3]. Although [14]C dates place RT5 within the LNBA time range (Fig. 2c, Supplementary Table 6), it occupies an unexpected position in the phylogeny, appearing further derived along branch 0 and being part of a polytomy that gave rise to three independent lineages (Fig. 2b). We explored the possibility of this polytomy reflecting a limited phylogenetic resolution at that node, given the exclusion of missing data (complete deletion) in our analysis. By contrast, we were able to replicate this topology after inclusion of all data in our phylogenetic reconstruction (Supplementary Fig. 3). In addition, since the coverage of RT6 was too low for confident SNP-calling, its genotype was manually explored after clipping 2 bp from the 3′ and 5′ ends of all reads to avoid the interference of post-mortem damage with our SNP assignments (see Methods). As RT5 possesses five unique, non-homoplastic, SNPs, we assessed their similarity with RT6. For all such positions covered in RT6, it possesses the identical SNP genotype as RT5 (Supplementary Table 7), suggesting that the two strains are likely identical, or are at minimum closely related. Such a result is expected since the two individuals derive from a double burial (Fig. 1a).

We further assessed the relatedness of RT5 with the previously published Iron Age isolate (RISE397)[2]. The coverage of RISE397 was too low (0.25-fold) to permit robust phylogenetic analysis. Therefore, manual genotyping of phylogenetically "diagnostic" SNPs was performed to infer its possible positioning (see Methods). As 22.5% of the RISE397 genome was covered ≥1-fold, and only 3.7% of the genome was covered ≥2-fold, we considered all mapping reads for this analysis. Reads covering informative positions were authenticated based on whether they carried diagnostic SNPs or/and carried additional substitutions that were consistent with terminal deamination, which is characteristic of aDNA (Supplementary Data 3)[37]. RISE397 DNA reads cover 26% (12/46) of the diagnostic positions leading to the RT5 node, making it clearly distinct from the ancestral 0. PE2 and 0.PE7 genomes since it possesses the derived-state (CO92 reference) allele in positions where the 0.PE2 and 0.PE7 genomes have ancestral variants (Supplementary Data 3, Supplementary Fig. 4). In addition, 30% (7/23) of SNPs extending from RT5 to the Justinian 2148 branch were covered in RISE397 (Supplementary Data 4). All such positions were identical to RT5, exhibiting the ancestral alleles, and hence supporting that RISE397 is basal to lineage 0.ANT1 (Supplementary Data 3, Supplementary Fig. 4). Finally, only one of five private-derived RT5 SNPs was covered in RISE397, where it instead matched the reference sequence (Supplementary Table 7). Although achieving a higher coverage would be necessary to verify its precise positioning, our analysis suggests that RISE397 and RT5 are closely related strains and potentially originated from the same progenitor (Supplementary Fig. 4). The node which gave rise to RT5 and perhaps also RISE397 seems to have initiated a radiation event that gave rise to all historical and extant *Y. pestis* lineages that have been isolated to date, with the exception of the more basal 0.PE2, 0.PE7 and LNBA (Fig. 2b, Supplementary Fig. 4).

**Y. pestis divergence time estimates and demographic analysis.** To estimate the time to the most recent common ancestor (tMRCA) of all *Y. pestis* strains, we employed the coalescent constant size, and the coalescent Bayesian skyline models implemented in BEASTv1.8[38-40]. According to our marginal likelihood (MLE) estimates computed via path sampling (PS) and stepping stone sampling (SS)[41], the Bayesian skyline model is the more suitable demographic model for the current dataset (Supplementary Table 8 and Methods). Nevertheless, both analyses produced overlapping divergence date intervals (Supplementary Fig. 5). While the constant size method is unlikely to represent a realistic demographic history model for epidemic pathogens, it has often been a preferred dating method[2,3,12]. Here, it produced a coalescent date estimate of 6797y BP (HPD 95%: 5299–8743) for *Y. pestis*, which is about 2000 years older than the oldest strains thus far identified[2,3] (Supplementary Table 9). In contrast, the coalescent skyline method, which allows for population size changes through time, produced a narrower interval and a younger tMRCA estimated at 5727y BP (HPD 95%: 4909–6842) (Supplementary Fig. 6, Supplementary Table 9).

In addition, using the Bayesian skyline model, we estimated effective population size ($N_e$) changes across the evolutionary history of *Y. pestis*. Our skyline plot (Supplementary Fig. 7) reveals an initial population expansion at ~4000y BP. Such an increase corresponds temporally with the RT5 polytomy described here (Fig. 2b) that we date to 4011y BP (HPD 95%: 3760–4325) (Supplementary Fig. 6). Although such effect could arise as a result of sampling bias in the data, it is of note that from 159 modern *Y. pestis* strains considered for the present analysis, only 13.2% are phylogenetically ancestral to the described

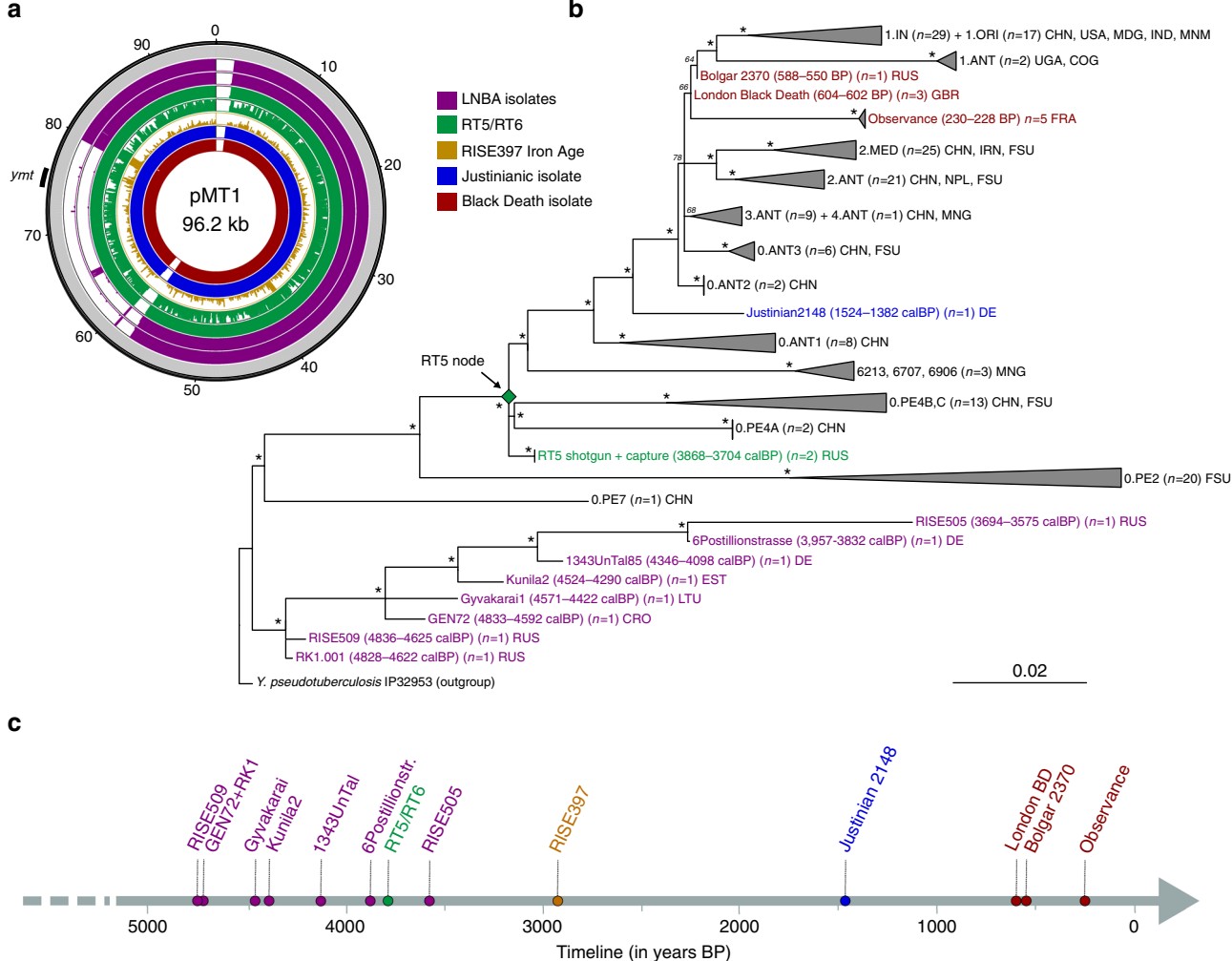

**Fig. 2** *Y. pestis* genomic characterisation and maximum likelihood phylogeny. **a** pMT1 coverage plots made with the Circos[80] software. The plots were constructed to a maximum coverage of three-fold, and average coverage was calculated over 100-bp windows. The presented strains are in the following order starting from the outmost: CO92 pMT1 (reference) in grey, the oldest (RISE509) and youngest (RISE505) isolates from the LNBA lineage are shown in purple, RT5 and RT6 are shown in green, the Iron Age RISE397 isolate is shown in brown, a Justinianic isolate from Altenerding (Germany) is shown in blue and a London Black Death isolate is shown in red. The position of the *ymt* gene within the pMT1 plasmid is indicated on the plot. **b** A worldwide dataset of *Y. pestis* ancient and present-day chromosomal genomes ($n = 179$) was used to reconstruct the phylogenetic tree, considering 1054 SNP positions (see Supplementary Fig. 3 for a phylogeny using all sites). The main branches were collapsed to enhance the clarity of the phylogeny, and branch lengths are shown as number of substitutions per site. The newly sequenced RT5 strain (green) was included in the phylogeny alongside eight Bronze Age strains belonging to the LNBA lineage (purple), a single Justinianic strain (blue), and nine second pandemic strains (red). Asterisks denote bootstrap values >95 (1000 bootstrap iterations carried out). The two-sigma (95.4%) radiocarbon or archaeological dates of Bronze Age and historical strains are shown. Country or geographical region abbreviations are as follows: CHN (China), USA (United States of America), MDG (Madagascar), IND (India), IRN (Iran), MNM (Myanmar), RUS (Russia), GB (Great Britain), DE (Germany), FRA (France), MNG (Mongolia), NPL (Nepal), FSU (Former Soviet Union), CGO (Congo), UGA (Uganda), LTU (Lithuania), EST (Estonia) and CRO (Croatia). See also Supplementary Fig. 4 for the inferred phylogenetic positioning of RISE397. **c** Timeline spanning radiocarbon and archaeological dates, from which *Y. pestis* genomic data have been included in this study. Points on the timeline indicate median dates

polytomy, while the majority derive from it (Fig. 2b). Subsequently, our skyline plot reveals a population decline starting at ~300y BP that is immediately followed by an increase (Supplementary Fig. 7). This result is likely associated with the coalescence times of the most extensively sampled modern isolates, particularly those related to the third plague pandemic (Fig. 2b, Supplementary Fig. 6).

***Y. pestis* virulence factor analysis.** The presence of several *Y. pestis* virulence-associated genes was evaluated in RT5 (Fig. 3). While the LNBA, 0.PE2 and 0.PE4 strains seem to lack certain virulence determinants, RT5 harbours all known virulence factors

with the exception of the filamentous prophage (YpfΦ), which is, however, most consistently identified among 1.ORI strains (Fig. 3, Fig. 2b)[42,43].

Another important gene for *Y. pestis* virulence is *pla* located on the species-specific pPCP1 plasmid[44]. Although the gene is largely conserved among *Y. pestis* strains, an isoleucine (ancestral) to threonine (derived) alteration at amino acid position 259 has been used to differentiate the most basal isolates (LNBA, 0.PE7, 0.PE2 and 0.PE4) from the rest of *Y. pestis*[44]. In RT5, the *pla* genotype was manually explored, and was found to exhibit the ancestral allele (Supplementary Fig. 8). Although the ancestral *pla* allele has been associated with a less-efficient bacterial dissemination in mammals[44], modern strains from lineages 0.PE4 and 0.

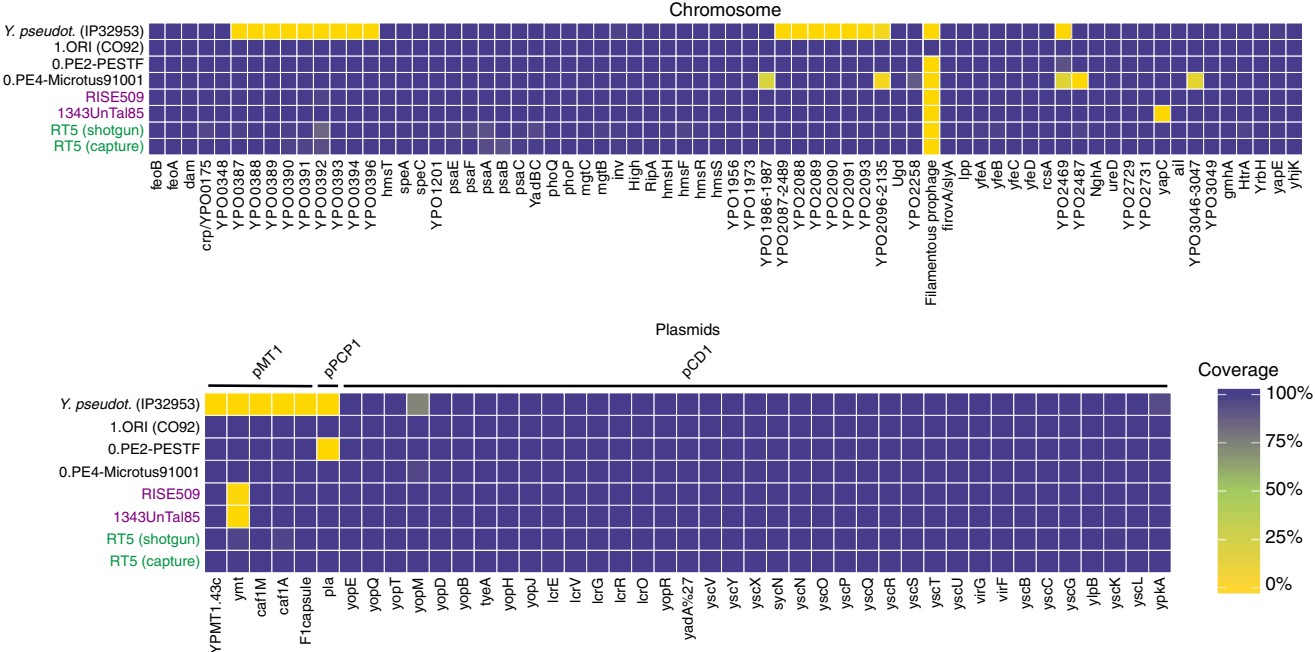

**Fig. 3** Heat map of coverage across virulence-associated genes. The virulence potential of RT5 shotgun-sequenced and captured genomes is compared to representative strains from the LNBA lineage, namely, RISE509 (whose virulence profile is identical to that of isolates RK1.001, GEN72, Gyvakarai1 and Kunila2[3]) and 1343UnTal85 (whose virulence profile is identical to 6Postillionstrasse and RISE505[3]). In addition it is compared to modern-isolate representatives 0.PE4-Microtus91001, 0.PE2-PestoidesF and 1.OR1-CO92, and to *Y. pseudotuberculosis* (strain IP32953). The virulence factors inspected were located on the *Y. pestis* chromosome, as well as on the pMT1, pPCP1 and pCD1 plasmids. The percentage of each gene covered (scale bar) was computed and plotted in the form of a heatmap using the ggplot2[79] package in R

PE7 have proven to be potent inducers of bubonic plague in humans[12].

In addition, we manually explored the status of *ureD*, PDE-2, PDE-3 and *rcsA*, all of which have been either lost or inactivated in *Y. pestis* by substitution or single-nucleotide InDels. The inactivation of these genes contributes to *Y. pestis*' ability to colonise, block and be transmitted via fleas (for more details, see Methods)[19,45]. Their active variants have been identified in previously published LNBA strains[2,3], thus suggesting either an inability or a lower efficiency in arthropod-based transmission. By contrast, we find that RT5 possessed the inactive form of all those genes, with the exception of a nonsense mutation in PDE-3 where it shows the ancestral allele (Supplementary Fig. 9). Together with the active *ymt* locus on the pMT1 plasmid (Fig. 2a), this suggests that RT5 was already adapted to the flea vector during the Bronze Age. Moreover, immune evasion by suppression of flagellar genes in *Y. pestis* is considered an important evolutionary advantage associated to a more complex niche adaptation that is absent in its closest ancestor *Y. pseudotuberculosis*[46]. The *flhD* regulatory gene is part of the *flhDC* operon and is expressed in a temperature-dependent manner in *Y. pseudotuberculosis*[46], but is inactive in all extant and historical *Y. pestis* strains sequenced to date. Although the strains belonging to the LNBA lineage encompass the active variant of *flhD*, RT5 contains the derived, inactive form (Supplementary Fig. 9).

As presented in the phylogeny (Fig. 2b), RT5 appears closely related to lineage 0.PE4 also referred to as "microtus". Since certain "microtus" strains have been associated with a decreased pathogenicity in humans, we inspected genes previously identified as responsible for this phenotype and verified their status in RT5 (Supplementary Table 10)[47,48]. Some of these loci seem to have been lost in 0.PE4 (Fig. 3 and Supplementary Table 10), and others have been disrupted by insertion sequences (IS) (*IS100* and *IS285*) or inactivated by substitutions/InDels (Supplementary

Table 10). In RT5, all such genes appear in their active form (Fig. 3, Supplementary Fig. 10), and therefore we have no evidence that this strain had a decreased virulence in humans.

## Discussion

Our results contribute to investigations regarding the evolution of *Y. pestis* and its disease potential in past human populations. We used shotgun sequencing and in-solution capture to reconstruct *Y. pestis* genomes from Bronze Age individuals (RT5 and RT6) in the Samara region. In addition, we retrieved a 4.2-fold human genome from individual RT5 through shotgun sequencing. Population genetic analysis identified individual RT5 as having close genetic affinity to EBA European populations and MLBA populations from the Eurasian steppe region (Fig. 1b). In particular, we show the presence of Yamnaya-related as well as farmer-related ancestry in RT5 (Fig. 1c). Our genomic characterisation is in line with previously described BA Eurasian populations, including "Srubnaya" individuals from Samara[26], where European farmer-related ancestry becomes present in Central Asia during the MBA as a result of population movements from Europe back into the steppe region[24,26].

We identify two ~3800-year-old individuals (Fig. 1a, Supplementary Table 3) out of nine analysed that were infected with *Y. pestis* at the time of their deaths. The Samara *Y. pestis* genomes presented here reveal greater lineage diversity during the Bronze Age than was previously described. Compared to the recently published LNBA isolates[2,3], RT5 and RT6 form a distinct branch in the *Y. pestis* phylogeny (Fig. 2b), deriving from a polytomy that gave rise to at least three separate lineages, two of which have persisted to the present day. RT5 falls only five derived SNPs away from the described polytomy (Supplementary Table 7). Our dating analyses consistently suggest the presence of this putative ancestor at ~4000y BP (Supplementary Fig. 6)

followed by a population expansion shortly after that time (Supplementary Fig. 7). Though its place of origin is not yet empirically identified, given the close genetic and temporal affinity to RT5 (Fig. 2b), a steppe source is plausible. Given that previous research has proposed a relationship between rapid *Y. pestis* expansions and historical plague epidemics in humans[12], future investigations of lineage diversity from modern and ancient sources may reveal additional details on this ancient radiation event.

Apart from RT5, a second known lineage that emerged as part of the described polytomy is "microtus" (0.PE4)[12,49]. 0.PE4 is found today in Central and East Asia, with some isolates being associated with bubonic plague infections in humans[12], and others being characterised as avirulent to humans[47,48,50,51]. Despite the variation in pathogenicity of 0.PE4 isolates, the phylogenetically related RT5 strain seems unaffected by genetic alterations/disruptions associated with reduced virulence (Fig. 3, Supplementary Fig. 10 and Supplementary Table 10). In addition, the third and most diverse branch established during this radiation event is one that gave rise to a multitude of *Y. pestis* lineages, which survive to the present day (Fig. 2b). These include 0.ANT along with the strain that caused the first historically documented plague pandemic (Plague of Justinian––sixth century)[5,6]; the entire branch 1 that includes strains responsible for the second (Black Death—fourth century)[4,8,36] and third (China—19th century) plague pandemics[11] and branches 2, 3 and 4 that are typically isolated from modern sylvatic rodents in Central and East Asia (Fig. 2b)[11,12].

A recent study has suggested that flea-adapted *Y. pestis*, along with its potential to cause bubonic plague in humans, likely originated around 3000y BP[2]. Contrary to such conclusions, the lineage giving rise to our *Y. pestis* isolates (RT5 and RT6) likely arose ~4000 years ago (Supplementary Tables 6 and 9), and possessed all vital genetic characteristics required for flea-borne transmission of plague in rodents, humans and other mammals. These include a fully incorporated *ymt* locus (Fig. 2a), and the inactive forms of *ureD*[52], PDE-2, PDE-3 and *rcsA* genes[19], as well as an inactive *flhD* flagellin gene (Supplementary Fig. 9). Moreover, our analysis of the previously published Iron Age RISE397 strain from modern-day Armenia[2] revealed its close relationship to RT5 and RT6 (Supplementary Fig. 4). Note that the modern 0.PE2 and 0.PE7 lineages, which are known to possess all genomic characteristics that confer adaptation to fleas[19], fall ancestral to RT5 (Fig. 2b) and RISE397 (Supplementary Fig. 4), but are more derived than the LNBA lineage. Our phylogenetic and dating results thus suggest that 0.PE2 and 0.PE7 also originated during the Bronze Age, with their mean divergence here estimated to 4474 (HPD 95%: 3936–5158) and 5237 (HPD 95%: 4248–6346) years BP, respectively, based on the Bayesian skyline model (Supplementary Table 9). While these lineages may have been confined to sylvatic rodent reservoirs during the EBA, the possibility that they co-circulated among human populations contemporaneously with the LNBA lineage should be considered. Although the places of origin of 0.PE2 and 0.PE7 are not known, today, their strains are isolated from modern-day China and the Caucasus region. In terms of their disease potential, both 0.PE2 and 0.PE7 possess pMT1 plasmids with fully functional *ymt* genes, but 0.PE2 strains lack pPCP1[44], and though frequently recovered from sylvatic rodent reservoirs, their virulence in humans is not known. On the other hand, the more basal 0.PE7 contains pPCP1[2] and has previously been associated with human bubonic plague[12]. It is, therefore, tempting to hypothesise that efficient flea adaptation in *Y. pestis*, as well as the potential for bubonic disease, might have evolved earlier than 5000 years ago.

Overall, the detection of *Y. pestis* in Bronze Age human remains from Eurasia has suggested the presence of the pathogen in this vast geographic area along with its ability to cause bubonic plague millennia before the first historically documented plague pandemic[2,3]. It seems possible that already in the Bronze Age, with the establishment of transport and trade networks, the interconnectivity between Europe and Asia that is also reflected in the ancient human genomes, likely contributed to the spread of infectious disease. Similarly, the abundant trade routes of the medieval period are considered the main conduit for plague's movement between Asia and Europe[8,12]. Our current data suggest a more complex model, where at least two human-associated lineages (LNBA and RT5) with different transmission potentials were established in Eurasia during the Bronze Age (Fig. 2b, c). Whether these lineages had equal prevalence among human populations, and the extent to which human practices contributed to their dissemination, are concepts requiring further investigation. Additional Bronze Age/Iron Age genomes could provide further insights into the early stages of *Y. pestis* evolution, and help pinpoint key events that contributed to the high virulence and spread of one of humankind's most notorious pathogens.

## Methods

**Sampling and extraction**. All laboratory procedures were performed in the dedicated ancient DNA facilities of the Max Planck Institute for the Science of Human History in Jena, Germany.

Teeth from nine individuals (one tooth from each), buried in the Mikhaylovka II tombs of the Samara region in Russia, were sectioned in the cementoenamel junction using a coping saw and 50–100 mg of dental pulp was removed from each tooth using a dental drill.

Extraction of 50–60 mg of dental pulp from each tooth sample was performed using a previously described protocol; optimised for the recovery of short DNA fragments, most typical of ancient DNA[53]. An initial lysis step was performed over a 12–16 h incubation of the dental pulp powder in 1 ml of extraction buffer (0.45 M EDTA, pH 8.0, and 0.25 mg ml$^{-1}$ proteinase K) at 37 °C. Following extraction, DNA was bound to a silica membrane using a binding buffer containing guanidine hydrochloride (protocol previously described in ref. [53]) and purified in combination with the High Pure Viral Nucleic Acid Large Volume Kit (Roche). DNA was eluted in 100 μl of TET (10 mM Tris-HCl, 1 mM EDTA, pH 8.0 and 0.05% Tween20). One extraction blank and one positive extraction control (previously assessed cave-bearing specimen) were taken along for the extraction slot.

**Illumina library preparation and sequencing**. To screen all samples for the presence of *Y. pestis* and human endogenous DNA, 10 μl of each extract was converted into double-stranded Illumina NGS libraries, using a previously described protocol[54], without initial uracil-DNA-glycosylase (UDG) treatment[55]. A positive control (cave-bearing specimen) and a negative library control (H$_2$O) were taken along for the experiment. A total of 1 μl from each library was subsequently quantified using IS7/IS8 primers. A combination of two unique indexes (8 bp length of each index sequence) also containing the universal IS5/IS6 priming sites was assigned to each sample for subsequent multiplex sequencing[56]. The libraries were then indexed through a ten-cycle amplification reaction using the *Pfu Turbo Cx Hotstart DNA Polymerase* (Agilent). Indexed PCR products were purified using a Qiagen MinElute kit (Qiagen), eluted in TET (10 mM Tris-HCl, 1 mM EDTA, pH 8.0 and 0.05% Tween20) and then qPCR quantified using IS5/IS6 primers, to assess the efficiency of the indexing reaction. After this, indexed libraries were amplified for different amounts of cycles, to achieve a total of 10$^{13}$ DNA copies per reaction in order to avoid polymerase saturation and heteroduplex formation. PCR products were again purified using a Qiagen MinElute kit (Qiagen) and eluted in TET (10 mM Tris-HCl, 1 mM EDTA, pH 8.0 and 0.05% Tween20). The concentration (ng μl$^{-1}$) of the indexed and amplified libraries was then measured using a 4200 Agilent Tape Station Instrument (Agilent). Finally, all samples were diluted and pooled at equimolar ratios to achieve a final 10 nM pool that would serve as template for sequencing.

**In silico screening for *Y. pestis* reads**. The sample pool was single-read sequenced on a HiSeq 4000 platform using a $1 \times 76 + 8 + 8$ cycles chemistry kit according to the manufacturer's protocol, to produce between 5,969,436 and 8,215,620 raw demultiplexed reads per sample. Pre-processing of reads was performed using the automated pipeline EAGER v1.92[57] to clip adaptors (using ClipAndMerge) and to filter reads for sequencing quality (minimum base quality 20) and length (keeping all reads ≥30 bp). Mapping was performed using BWA[58] implemented in EAGER to *Y. pestis* CO92 (NC_003143.1), using a –n parameter of 0.01, a –l seedlength of 16 and subsequently using SAMtools to filter for reads with

a mapping quality (−q) of 37. The MarkDuplicates tool in Picard (1.140, http://broadinstitute.github.io/picard/) was used to remove duplicates.

In addition, the Megan ALignment Tool (MALT)[27] was used to assess the metagenomic composition of the samples, as well as a screening tool for the identification of *Y. pestis*. All bacterial genomes available at GenBank were used as a reference database for the programme (NCBI RefSeq, December 2015). Pre-processed reads were used as input for MALT (version 0.3.6), and the parameters were set to 85 for the minimum percent identity (--minPercentIdentity), 0.01 for the minimum support parameter (--minSupport), using a top percent value of 1 (--topPercent) and the semi-global alignment mode. All the remaining parameters were set to default. The results were viewed in MEGAN6[59]. Putatively positive *Y. pestis* samples were evaluated by comparing the amount of reads mapping to *Y. pestis* CO92 (NC_003143.1) to the reads assigned by MALT on the *Y. pestis* and *Y. pseudotuberculosis complex* nodes (Supplementary Table 1).

**In-solution *Y. pestis* capture and deep-shotgun sequencing.** Rich double-stranded DNA libraries were prepared for in-solution capture and deep-shotgun sequencing of putatively positive *Y. pestis* samples, using 50 μl of extract (or 2 × 25 μl of extract), according to a previously described protocol[54], with an initial partial-UDG treatment step[60], where UDG in combination with endonuclease VIII (USER enzyme, New England Biolabs) were used to remove all deaminated cytosines (uracils) with the exception of terminal uracil nucleotides that lack 5′ phosphate. Double-indexing and subsequent library amplification steps were carried out as mentioned in the previous section "Illumina library preparation and sequencing". At this stage, the sample RT5 was diluted to 10 nM for deep-shotgun sequencing on a HiSeq 4000 platform using a 1 × 76+8+8 cycles chemistry kit. In addition, 1–2 μg of samples RT5 and RT6 were in-solution captured as described previously[3], where a combination of the following *Y. pestis* and *Y. pseudotuberculosis* genomes were used as templates for probe design: CO92 chromosome (NC_003143.1), CO92 plasmid pMT1 (NC_003134.1), CO92 plasmid pCD1 (NC_003131.1), KIM 10 chromosome (NC_004088.1), Pestoides F chromosome (NC_009381.1) and *Y. pseudotuberculosis* IP32953 chromosome (NC_006155.1). Samples were captured in separate wells of a 96-well plate, whereas extraction and library blanks (data not shown) with non-overlapping index combination were pooled and captured in a single well. Sequencing was performed on a HiSeq 4000 platform using both single-end (1 × 76+8+8 cycles) as well as paired-end (2 × 76+8+8 cycles) chemistry kits.

**Y. pestis read authentication and genome reconstruction.** Sequencing resulted in up to 1,140,960,213 raw reads per sample. Adaptor trimming of raw, demulti-plexed, reads was performed using leeHom[61]. Subsequently, length and quality-filtering steps were performed in EAGER, as mentioned in the previous section "In silico screening for *Y. pestis* reads". After pre-processing, captured paired-end and single-end reads from the same individuals were merged into a single file for mapping. BWA[58] integrated in EAGER was used for mapping against the *Y. pestis* CO92 reference (NC_003143.1)[62] using the following parameters: −n 0.1, −l 32, and subsequently SAMtools was used to filter for reads with mapping quality of 37 (−q 37). Mean coverages were estimated using QualiMap v.2.2.1[63] and DNA deamination profiles typical of aDNA were calculated using MapDamage 2.0[64]. For genome reconstruction, and for downstream SNP calling, the same pipeline was followed with a single alteration: after adaptor trimming, reads were inputted into EAGER and 2 bp were trimmed from each end using ClipAndMerge prior to length filtering and mapping to eliminate post-mortem damage that might affect downstream SNP calling.

**Read-length comparison of capture and shotgun *Y. pestis* reads.** Two datasets derived from the same individual (RT5), sequenced using the 1 × 76+8+8 cycles kit parameters, were used to compare the read-length distributions of shotgun-sequenced reads and captured reads. For this analysis, datasets were limited to the same genomic coverage (~9-fold), to ensure uniform comparison and avoid any biases that might arise from unequal coverage. Reads shorter or equal to 74 bp were considered for the analysis, to avoid the incorporation of reads that still contain traces of adaptor, or are longer than 76 bp. Box-plot comparisons and Student's *t*-test were calculated using R version 3.4.1[65].

**Y. pestis SNP calling and phylogenetic analysis.** For SNP calling, we used the UnifiedGenotyper of the Genome Analysis Toolkit (GATK)[66]. The newly produced RT5 shotgun-sequenced and captured genomes were analysed alongside 177 previously published *Y. pestis* genomes (179 in total), including one previously published historical strain from the Plague of Justinian[6], nine genomes from the second plague pandemic[4,8,36], eight previously published LNBA genomes[2,3] and a global dataset of 159 modern *Y. pestis* genomes (Supplementary Data 2)[11,12,14,15,47,62,67–70]. A *Y. pseudotuberculosis* strain (IP32953)[18] was used as an outgroup. A vcf file was produced for every sample using the "EMIT_ALL_SITES" in GATK[66], which generated a call for all positions in the reference genome. In addition, the custom Java programme MultiVCFAnalyzer v0.85[71] (https://github.com/alexherbig/MultiVCFAnalyzer) was used to produce a combined SNP table, including all variable positions across our dataset, with the exclusion of previously defined noncore regions and homoplasies, as well as repeat regions, tRNAs, rRNAs and tmRNAs[11,12]. In addition, we used ClonalFrameML[72] to identify additional

homoplasies or recombinant regions in our dataset by using a full-genome align-ment and a RAxML[73] ML SNP tree as input, as well as the -em and the -ignor-e_incomplete_sites options for running the programme. Through this analysis, we identified seven additional regions (resulting in 26 SNPs) and two homoplastic SNPs (28 SNPs in total), which were also excluded from the comparative SNP analysis (Supplementary Table 11). For the remaining data, SNPs were filtered according to the following criteria: (1) homozygous SNPs and reference alleles were called when covered at least three-fold with a minimum genotyping quality of 30, (2) in cases of heterozygous positions, a SNP or reference base was called when supported by at least 90% of the reads covering the respective position and (3) if none of the criteria could be fulfilled, a "N" was inserted in the respective position. A total of 3821 SNP positions were called in the current dataset.

From the resulting SNP alignment, two ML phylogenetic trees were inferred with RAxML (version 8.2.9)[73], using the generalised time-reversible (GTR) substitution model with gamma-distributed rates (six rate categories). The first included all data. For the second tree reconstruction, all columns with missing data were excluded (complete deletion), which resulted in a total of 1054 SNP positions to be considered for the phylogeny. A total of 1000 bootstrap replicates were carried out to estimate the topology support of each tree.

In addition, the phylogenetic positioning of RT6 (present study) and RISE397[2] was manually explored using the following methods:

- For RT6: To check whether RT5 and RT6 form the same phylogenetic branch, SNPs specific to the RT5 genome (*n* = 5, Supplementary Table 7) were assessed for their presence in RT6. RT6 possesses four out of five SNP positions covered at least one-fold. All positions covered encompass identical alleles to RT5.

- For RISE397: The SNP table produced by MultiVCFAnalyzer was filtered for all diagnostic positions leading from the root of the tree towards the RT5 node (*n* = 46 SNP positions). In addition, the SNP table was independently filtered for all positions leading from the RT5 node to the Justinianic node (branch represented by Justinian 2148 strain) (*n* = 23 SNP positions), for which RT5 appears to have ancestral alleles. Missing data (N's) were excluded from this SNP analysis. The state of all alleles in RISE397 was then manually inspected using the Integrative Genomics Viewer[74]. The reads covering all respective positions were visually authenticated by assessing whether they include terminal substitutions that could be explained by aDNA damage (Supplementary Data 3, 4).

**Divergence date and demographic analyses.** TempEst v1.5[75] was used to assess for the presence of temporal signal in the dataset. Inclusion of the ¹⁴C or archaeological dates for all ancient isolates resulted in a 0.6 correlation coefficient, which permitted the proceeding with molecular dating analysis. The software package BEAST v1.8[39] was used to estimate the divergence time of *Y. pestis* lineages, using the coalescent constant size[40] and the coalescent Bayesian skyline[38] models. The SNP alignment including only *Y. pestis* strains was used as input, after removal of all missing data (complete deletion). The tip dates of all modern *Y. pestis* strains were set to 0 years before present (BP). The ages of all ancient and historical *Y. pestis* genomes were estimated with their prior uniform distributions based either on the two-sigma (95.4%) ¹⁴C date interval[2,3,6] or the archaeological dates[4,8,36] in years BP, as follows: RISE509 (4836–4625, median: 4729), RK1.001 (4828–4622, median: 4720), GEN72 (4833–4592, median: 4721), Gyvakarai1 (4571–4422, median: 4485), Kunila2 (4524–4290, median: 4427), 1343UnTal (4346–4098, median: 4203), 6Post (3957–3832, median: 3873), RT5 (3868–3704, median: 3789), RISE505 (3694–3575, median: 3635), London Black Death (602–604, median: 603), observance (228–230, median: 229), Bolgar 2370 (550–588, median: 569) and Justinian 2148 (1382–1524, median: 1453). In addi-tion, we used MEGA7[76] to test whether there is an equal evolutionary rate across our phylogeny. The strict clock rate was rejected, and therefore we applied a lognormal relaxed clock[77] for all dating analyses, along with the GTR model of nucleotide substitution (six gamma categories), as previously performed[2,3]. An ML phylogeny was reconstructed using RaxML[73] and was used as a starting tree. The LNBA lineage and the rest of *Y. pestis* strains were constrained to be two separate monophyletic groups in BEAUti v1.8[39]. A single chain of 300,000,000 states was run for each model setup, sampling every 10,000 states. In addition, we estimated MLE using PS/SS sampling[41] as part of the same set-up in BEAUti to assess which of the two demographic models is best fit for our data. We run this analysis for an extra 300,000,000 states divided between 100 steps (3,000,000 states each), using an alpha parameter of 0.3. After run completion, the molecular dating results were viewed in Tracer v1.6 (http://tree.bio.ed.ac.uk/software/tracer/) to ensure that all expected sample sizes were above 200. TreeAnnotator was used to produce a maximum clade credibility tree with a 10% burn-in (excluding the first 3000 trees), which resulted in processing of 27,001 trees for each analysis with a Jeffreys prior distribution (1.0) for the population sizes. In addition, for the coalescent skyline analysis, we used 20 as the dimension for the population and group sizes. Once the chain was complete, we used LogCombiner[39] to resample MCMC states at lower frequency (every 300,000) with a 10% burn-in, and the resultant .log and .tree files were used as input for the skyline plot construction in Tracer v1.6 (http://tree.bio.ed.ac.uk/software/tracer/).

**Analysis of virulence factors**. As map-quality filtering may influence read mappability in certain chromosomal and plasmid regions, we used a mapping quality filter of 0 (–q parameter) to evaluate the presence or absence of chromosomal and plasmid virulence-associated genes in RT5 compared to previously published modern and LNBA *Y. pestis* genomes[48]. Bedtools[78] were used to calculate the percentage of gene covered across each region, and a heatmap was plotted using the ggplot2[79] package of R version 3.4.1[65].

In addition, the virulence-associated genes *flhD*, PDE-2, PDE-3, *ureD* and *rcsA*, which are known to have become inactivated in *Y. pestis* by either mutation or single-nucleotide insertions/deletions[19], were instead manually explored using IGV[74]. Gene *flhD*, associated with flagellar biosynthesis and whose silencing contributes to immune evasion, is inactivated by a frameshift caused by a T insertion, present at position 1,892,659 in CO92[46]. PDE-2, a phosphodiesterase gene contributor in biofilm degradation is inactivated by a T insertion in a six-T stretch at position 1,434,044 in CO92. In addition, PDE-3, also part of the same biofilm-degradation mechanism, is affected by two mutations, a C > T change (also called the PDE-*pe*' allele[19]), and a nonsense G > A substitution, which are, respectively, shown at positions 3,944,166 and 3,944,534 in *Y. pseudotuberculosis* IP32953[18]. The urease enzyme, *ureD*, that causes toxicity in fleas, is inactivated in *Y. pestis* by a G insertion in a six-G stretch, shown at position 2,997,296 in CO92. Finally, the *rcsA* gene, a component of the Rcs system that functions as an inhibitor to biofilm formation, is known to have become inactivated in *Y. pestis* by a 30 bp internal duplication, previously described in the strain KIM (NC_004088.1)[19].

In addition, as certain *Y. pestis* strains present in the closely related 0.PE4 "microtus" lineage have been previously associated to a reduced pathogenicity, genes associated to this attenuated phenotype[48] were explored in RT5, in relation to 0.PE4. The following regions were integrated into the presence/absence heatmap analysis described previously, as they are absent in microtus: YPO1986 to YPO1987, YPO2096 to YPO2135, YPO2469, YPO2487 to YPO2489 and YPO3046 to YPO3047 (region annotations are given as they appear in CO92) (Fig. 3). Additional genes, which appear to have been disrupted by IS elements, or by mutations/InDels (Supplementary Table 9), were visually inspected in IGV (Supplementary Fig. 10)[74].

**Data availability**. Raw sequencing data of the deep-sequenced RT5 and RT6 isolates have been deposited into the European Nucleotide Archive under project accession number PRJEB24296. Other data supporting the findings of the study are available in this article and its Supplementary Information files, or from the corresponding authors upon request.

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

## Acknowledgements

We thank Cosimo Posth, Marcel Keller, Michal Feldman and Wolfgang Haak for useful insights to the manuscript, as well as Alexander Immel and Stephen Clayton for computational support. In addition, we are thankful to Guido Brandt, Antje Wissgott and Cäcilia Freund for laboratory support. M.A.S., A.H., K.I.B. and J.K. were supported by the ERC starting grant APGREID, and by the Max Planck Society. C.C.W. was supported by the Max Planck Society and the Nanqiang Outstanding Young Talents Program of Xiamen University. D.K. was supported by a Marie Heim-Vögtlin grant from the Swiss National Science Foundation.

## Author contributions

M.A.S., R.I.T., K.I.B. and J.K. designed the study; R.I.T., V.V.K., V.A.T. and A.K. provided access to human archaeological material; M.A.S. and R.I.T. performed laboratory work; M.A.S., C.C.W., A.A.V., A.K.L., D.K. and A.H. performed data analyses; and M.A.S., K.I.B. and J.K. wrote the manuscript with input from all co-authors.

## Additional information

**Competing interests:** The authors declare no competing interests.

