## [Peer Review File · Nature Communications]

Reviewers' comments:

Reviewer #1 (Remarks to the Author):

Nature Communications manuscript
Spyrou et al.

Two highly interesting and important new discoveries are reported in this study. One is the occurrence of *Y. pestis* having all the characteristic genetic attributes consistent with the modern flea-borne transmission route as far back as the Bronze Age. A previous study uncovered strains from the same time period but which lacked the genetic signatures of efficient flea-borne transmission and speculated that flea-borne transmission arose much later. The present ms proves otherwise and indicates that different strains of *Y. pestis* were circulating during the Bronze Age. A second insightful finding is the population genetic analyses that place two modern, flea-borne isolates (O.PE2 and O.PE7) into a lineage branch that coincides with or even precedes the Bronze Age strain characterized here.

The ms is very well written, the data are very strong, the conclusions are well-supported by the analyses, and the results well-discussed in relation to previous studies. My only suggestion is that the two O.PE strains be included on the timeline shown in Fig. 2c.

Reviewer #2 (Remarks to the Author):

This manuscript by Spyrou and colleagues reports the sequence of a ~3,800-year old *Yersinia pestis* genome that carries the virulence plasmids of modern plague strains capable of infecting both mammals and arthropods, and thus likely capable of mounting epidemics of bubonic plague.

This genome is around 1,000 years older than the previously oldest sequenced bubonic plague isolate. This is an interesting observation. Though, it is not obvious to me how much it forces us to revise our understanding of the evolution of *Y. pestis* and the history of plague. Below, please find a list of questions and concerns that arose when I read the manuscript.

Major Comments

Age of bubonic plague: The authors claim that their results based on this new genome "suggest that the full ability for flea-mediated transmission causing bubonic plague evolved more than 1,000 years earlier than previously suggested". I believe this statement is not strictly accurate. I appreciate that an ancient genome with an associated C14 date represents harder evidence than the phylogenetically inferred Time to the Most Recent Common Ancestor (TMRCA) of all *Y. pestis* isolates likely to cause bubonic plague (*b_pestis*). That said, the authors are referring to the TMRCA of *b_pestis* isolates here. Their *b_pestis* TMRCA is actually not older than the one reported in Rasmussen et al. 2015, Cell (which admittedly comes with no associated confidence intervals). If the authors wish to make this claim, they should ideally demonstrate that the TMRCA of *b_pestis* is 1,000 years (or at least significantly) older in a phylogenetic reconstruction including the new isolate, than for a phylogeny without this isolate.

Phylogenetic analyses: The phylogenetic analyses do not always seem adequate. As the paper comprises only limited information on how these were implemented, I do not feel I can really comment on the specifics at this stage. However, for the work to be publishable (and reproducible) a number of points need to be addressed. These include:

- What is the temporal signal in the data, the absolute minimum requirement would be to include a correlation between dates and distance to the root.
- Why was a relaxed lognormal clock chosen in the model? This needs to be justified and other clock models need to be considered and tested.
- Different demographic models were considered. Was the fit of the different models tested? If not, this needs to be done, and the discussion of the results should focus on the best fitting model.
- The results from the Bayesian Coalescent Skyline (BCS) plot (FigS5) seem over-interpreted to me. BCS plots are a depiction of the density of coalescence events, which are very shallow for this dataset, so that a single sequence can create the type of “blips” that the authors discuss as evidence for some major demographic events in the history of the plague.
- Modern *Y. pestis* is considered as strictly clonal. Though, we do not know whether this was the case for its ancestors and its closest relative *Y. pseudotuberculosis* (used as an outgroup here) is anything but clonal. I recommend the authors test for the presence of homoplasies and may wish to deploy other approaches to detect genetic recombination, for instance using ClonalFrame.

Politomies: The authors discuss extensively a politomy dated to around 3,000bp. Polytomies can indicate rapid demographic expansion, or they can simply be caused by insufficient phylogenetic resolution. This polytomy of three branches, includes one that leads to the Angola (O.PE3) strain as the only tip. As the authors mention in the paper, Angola is a very atypical strain with no known close relative, an excess of private SNPs and an obscure origin, and it might even be a strain that has been passaged in a lab. The authors wrote in the discussion:

“In contrast to O.PE4, the isolation history and natural provenance of O.PE3 are not published, and since the only available genome representative of this lineage is atypical with a high number of private SNPs and a unique genome architecture (Fig. 2b), we have excluded it from further interpretations.”

Given the statement above that Angola was excluded from further interpretations, I was surprised to see it included in Figure 2b, and even more so for the authors to discuss at length a politomy involving the Angola strain (and that would not exist without the inclusion of Angola). If the authors have no confidence in the phylogenetic placement of Angola, I recommend they ignore any mention of this polytomy.

Minor Comments

Figure 2b: What do stars denote in figure 2b. The star symbol is sometimes used to indicate statistical support higher than 95%, but that is unlikely to be the case here, as bootstrap support values up to 99% are indicated in full on some nodes.

Write-up: The paper needs to make it clearer what are the factual results, the inferred results and the speculations.

Reviewer #3 (Remarks to the Author):

The oldest evidence for *Yersinia pestis* was found in several Late Neolithic-Bronze Age individuals across Eurasia in a previous work by Rasmussen et al. (2015), a finding that was suggestive of a previously unknown epidemic coming into Europe along with the steppe pastoralists at the end of the Neolithic. However, the lack of virulence associated plasmids in these ancient strains suggested that the Bronze Age epidemic could not trigger the most deadly form of the disease, the so-called "bubonic" form that is able to be transmitted to humans through bites from infected fleas. Therefore, it was yet not well understood when *Yersinia pestis* acquired its modern, high pathogenicity.

In this work the authors have found evidence of the pathogen in nine Late Bronze Age individuals from the Central steppe region (Samara, in Russia) and retrieved the complete genome of *Y. pestis* in two individuals by combining two different methods (plus the complete human genome of one of them). The analysis of the pathogen showed it already had the virulence factors and therefore, showed a more-derived position in the *Yersinia* phylogenetic tree than other contemporaneous samples. The analysis of the human genome (and also the Y chromosome) placed this individual along the steppe nomads that transformed the genetic European landscape by the end of the Neolithic. They also dated by coalescent methods the origin of the pathogenic lineage and found a date of at least 4,000 years ago, 1,000 years older than previously assumed because of its absence in previous Bronze Age samples. In conclusion, this work proves that both *Yersinia* lineages existed contemporaneously along the Central Asian steppes.

I found this an interesting paper that solves the obvious question of the date of the pathogenic strain and that therefore can be of interest to researchers in different fields, from experts on infectious diseases to evolutionary biologists and archaeologists. The methodology used, both experimentally and computationally, are the appropriate ones. In addition, the paper is clearly written.

I don't have major concerns or criticisms. Maybe the authors could mention this paper: <https://www.biorxiv.org/content/early/2017/05/09/135962> in which there is a general analysis of hundreds of Late Neolithic and Bronze Age samples across Western Europe and here they found evidence of a 90 population replacement of the British Neolithic populations with the arrival, from The Netherlands (or from surrounding regions) of the Bell Beakers, something that is clearly a following up of the east-west expansion that brought the steppe ancestry into Eastern and Central Europe. In the light of their findings, it is likely that a pathogenic *Yersinia pestis* could contribute to the extreme social and demographic turn-over detected in this work.

Also, the authors mention that the *Yersinia* reads harbour the signs of post-mortem damage at the ends of the reads that are proxy of authenticity. These damage patterns correlate with the age of the sample (Sawyer et al. 2012) and the authors have now generated *Yersinia* reads from quite different periods, along the last 4,000 years. Could it be possible to create a Figure to see if the extent of damage pattern is correlated with temporal dates on the pathogen strains? I have seen some people expressing doubts on the paleopathogen studies, so an evidence of this line would be nice, although I am not sure if there is enough data.

We are grateful to all three reviewers for their useful suggestions and remarks and believe that our paper has improved as a result of their input. Below we address all comments through a point-by-point response.

Reviewer #1 (Remarks to the Author):

Nature Communications manuscript
Spyrou et al.

Two highly interesting and important new discoveries are reported in this study. One is the occurrence of *Y. pestis* having all the characteristic genetic attributes consistent with the modern flea-borne transmission route as far back as the Bronze Age. A previous study uncovered strains from the same time period but which lacked the genetic signatures of efficient flea-borne transmission and speculated that flea-borne transmission arose much later. The present ms proves otherwise and indicates that different strains of *Y. pestis* were circulating during the Bronze Age. A second insightful finding is the population genetic analyses that place two modern, flea-borne isolates (0.PE2 and 0.PE7) into a lineage branch that coincides with or even precedes the Bronze Age strain characterized here.

The ms is very well written, the data are very strong, the conclusions are well-supported by the analyses, and the results well-discussed in relation to previous studies. My only suggestion is that the two 0.PE strains be included on the timeline shown in Fig. 2c.

We thank Reviewer #1 for their encouraging feedback.

The timeline in Figure 2c is meant to represent the temporal range of individuals from which reconstructed ancient *Y. pestis* strains have been securely dated (either with radiocarbon dating, or using archaeological context information). Unfortunately, to our knowledge, there has not yet been an ancient strain reported for lineages 0.PE2 and 0.PE7. We would, therefore, prefer to avoid including their divergence estimates within the timeline since those represent extrapolated dates, with large higher posterior density intervals, that are subject to change once more ancient genomes or modern diversity are incorporated in future molecular dating attempts.

Reviewer #2 (Remarks to the Author):

This manuscript by Spyrou and colleagues reports the sequence of a ~3,800-year old *Yersinia pestis* genome that carries the virulence plasmids of modern plague strains capable of infecting both mammals and arthropods, and thus likely capable of mounting epidemics of bubonic plague.

This genome is around 1,000 years older than the previously oldest sequenced bubonic plague isolate. This is an interesting observation. Though, it is not obvious to me how much it forces us to revise our understanding of the evolution of *Y. pestis* and the history of plague. Below, please find a list of questions and concerns that arose when I read the manuscript.

Major Comments

Age of bubonic plague: The authors claim that their results based on this new genome “*suggest that the full ability for flea-mediated transmission causing bubonic plague evolved more than 1,000 years earlier than previously suggested*”. I believe this statement is not strictly accurate. I appreciate that an ancient genome with an associated C14 date

represents harder evidence than the phylogenetically inferred Time to the Most Recent Common Ancestor (TMRCA) of all *Y. pestis* isolates likely to cause bubonic plague (*b_pestis*). That said, the authors are referring to the TMRCA of *b_pestis* isolates here. Their *b_pestis* TMRCA is actually not older than the one reported in Rasmussen et al. 2015, Cell (which admittedly comes with no associated confidence intervals). If the authors wish to make this claim, they should ideally demonstrate that the TMRCA of *b_pestis* is 1,000 years (or at least significantly) older in a phylogenetic reconstruction including the new isolate, than for a phylogeny without this isolate.

We thank Reviewer #2 for their constructive comments. As part of our manuscript, we are attempting to expand on previous interpretations regarding the emergence of flea-adaptation in *Y. pestis*. Below are three points that we regard as favourable towards our proposed scenario:

1. Direct molecular evidence of flea-adaptation:

Rasmussen *et al.*¹ hypothesize that the potential for bubonic disease was not acquired by *Y. pestis* before 3,000y BP. Their support for this comes from the identification of necessary virulence factors in a low coverage (0.25x) Iron Age isolate (RISE397 dated to 2,900 BP), and their absence in isolates from earlier time periods. Here, we instead make use of additional direct evidence from two 3,800-year-old *Y. pestis* genomes (one of which we sequence to 32x coverage) that show genetic signatures of flea-adaptation, as well as existing data from extant lineages (0.PE2 and 0.PE7), to revisit this hypothesis. Our presented DNA evidence alone suggests that the potential for bubonic disease existed at least 1,000 years earlier than previously thought. To our knowledge, this is the earliest evidence of such a strain to-date, since the previously published ancient *Y. pestis* genomes from the Late Neolithic and Bronze Age time periods do not possess the genomic characteristics of flea-adaptation.

2. The date of RT5 pushes back the divergence time of its lineage by 1,000 years:

We agree with Reviewer #2 that our new dating analysis does not show a marked difference in the tMRCA of all *Y. pestis* compared to previously published estimates. However, the part of our analysis that refers to the proposed “birth-time of bubonic plague” does show marked differences to previous work. Such differences arise as a result of the RT5 addition in the molecular dating. Specifically, when attempting to estimate the divergence time of the RT5 lineage, we obtain a date of ~4,000 years BP with a narrow confidence interval (HPD 95%: 3,753-4,215). We believe that this estimate is accurate since our securely ¹⁴C dated RT5 isolate falls only 5 SNPs away from its origin. In previously published work by Rasmussen *et al.*, this node (therein represented as the one that gave rise to extant lineage 0.PE4) is dated to ~3,000 years ago, which also correlates with their proposed birth-time of bubonic plague (please refer to Figure S5 in Rasmussen *et al.*). Therefore, by our estimations, we indeed show that the “tMRCA of *b_pestis*”, as proposed by Rasmussen and colleagues, is at least 1,000 years older than previously reported.

3. Inference of bubonic plague emergence:

Rasmussen *et al.* implicitly suggest that only after the birth of the aforementioned lineages did *Y. pestis* acquire its bubonic disease potential, based the functionality of a single substitution located on the *pla* gene of *Y. pestis* (please refer to Figure 6 in the Rasmussen *et al.*, 2015 paper for an illustration¹). This substitution seems to be shared by all lineages more divergent than 0.PE4, including those that caused the 1st, 2nd and 3rd plague pandemics. However, to our knowledge, although this substitution enhances bacterial dissemination, it is not absolutely essential for bubonic disease, since strains belonging to 0.PE4 as well as the most basal modern lineage 0.PE7 (here dated to >5,000 years ago) have been isolated from human bubonic plague cases in China (see publication by Cui *et al.*, 2013²). In our manuscript we contrast the suggestion of a 3,000-year-ago emergence of bubonic plague and propose that, based on the existing evidence, it appears that the

bubonic disease potential in *Y. pestis* may have been acquired even earlier than 5,000 years ago, although such a hypothesis awaits support from additional aDNA data.

Phylogenetic analyses: The phylogenetic analyses do not always seem adequate. As the paper comprises only limited information on how these were implemented, I do not feel I can really comment on the specifics at this stage. However, for the work to be publishable (and reproducible) a number of points need to be addressed. These include:

- What is the temporal signal in the data, the absolute minimum requirement would be to include a correlation between dates and distance to the root.

We used TempEst v1.5³ to assess the temporal signal in our data, after including all dates from ancient/historical isolates. Our resulting correlation coefficient is 0.6, which suggests that the root to tip distance correlates with time.

This analysis is now mentioned in the Methods section of our manuscript as follows (lines 567-572):

“TempEst was used to assess for the presence of temporal signal in the dataset. Inclusion of the ¹⁴C or archaeological dates for all ancient isolates resulted in a 0.6 correlation coefficient, which allowed us to proceed with dating analysis.”

- Why was a relaxed lognormal clock chosen in the model? This needs to be justified and other clock models need to be considered and tested.

We tested clock rates using MEGA7⁴, where a strict clock was rejected with statistical significance. This analysis is now mentioned in the Methods of our manuscript as follows (lines 586-590):

“In addition, we used MEGA7 to test whether there is an equal evolutionary rate across our phylogeny. The strict clock rate was rejected, and therefore, we applied a lognormal relaxed clock for all dating analyses...”

- Different demographic models were considered. Was the fit of the different models tested? If not, this needs to be done, and the discussion of the results should focus on the best fitting model.

We have followed the reviewer’s suggestion and tested which is the best-supported model for our data by estimating marginal likelihoods using path sampling (PS) and stepping stone (SS) sampling⁵ in BEAST. Unfortunately, we came across a number of software errors when attempting to use path sampling in BEAST2. Therefore, we switched to BEASTv1.8⁶ for the entire molecular dating analysis where the implementation of PS and SS are standardized and easily applicable. For this, we omitted the Birth-Death Skyline⁷ (BD-Sky) model from our analysis, as this demographic model is not yet implemented in BEASTv1.8. In addition, given its similarity to the Bayesian Skyline model, a BD-Sky analysis is unlikely to provide us with additional insights for the current study. This is supported by our initial dating analysis where both models produced almost identical TMRCA estimates (mean estimates: 5,915 BP for Birth-Death Skyline and 5,825 for Coalescent Skyline using BEASTv2).

BEASTv1.8⁶ produced very similar dates compared to our previous estimates using BEASTv2⁸ (*Y. pestis* mean tMRCA using BEASTv2: 5,825y BP and using BEASTv1.8: 5,730y BP), which serves as an independent confirmation for our dating analyses. According to our marginal likelihood estimates, the Coalescent Bayesian Skyline model is the more favoured compared to the Coalescent Constant Size model for the present data (see Supplementary table 8). However, given the close similarity of the likelihood estimates produced for the two models, we have kept both tMRCAs within our manuscript (the new divergence dates produced by BEASTv1.8 can now be found in Supplementary table 9 and in the Main Text). In addition, we have now added a figure to demonstrate the overlapping

divergence date distributions produced by BEASTv1.8 for the two models (see Supplementary fig. 5).

- The results from the Bayesian Coalescent Skyline (BCS) plot (FigS5) seem over-interpreted to me. BCS plots are a depiction of the density of coalescence events, which are very shallow for this dataset, so that a single sequence can create the type of “blips” that the authors discuss as evidence for some major demographic events in the history of the plague.

Although we do see a correlation between an increased population size (Figure S5) and a radiation event in our phylogeny (Figure 2b), we recognise the reviewers concerns and have therefore acknowledged that sampling bias in our data could influence a population size analysis. We write: “Our Skyline plot (Supplementary fig. 7) reveals an initial population expansion at ~4,000y BP. Such an increase corresponds temporally with the RT5 polytomy described here (Fig. 2b) that we date to 3,962y BP (HPD 95%: 3,753-4,215) (Supplementary fig. 6, Supplementary Table 8). **Although such effect could arise as a result of sampling bias in the data**, it is of note that from 159 modern *Y. pestis* strains considered for the present analysis, only 13.2% are phylogenetically ancestral to the described polytomy, while the majority derive from it (Fig. 2b)”

- Modern *Y. pestis* is considered as strictly clonal. Though, we do not know whether this was the case for its ancestors and its closest relative *Y. pseudotuberculosis* (used as an outgroup here) is anything but clonal. I recommend the authors test for the presence of homoplasies and may wish to deploy other approaches to detect genetic recombination, for instance using ClonalFrame.

Previous studies, such as the one by Cui *et al*², have sufficiently demonstrated clonality in modern and historical *Y. pestis* lineages. Rasmussen *et al*.¹ and Andrades-Valtueña *et al*⁹ have shown that the most basal lineage of *Y. pestis*, also referred to as LNBA, does not seem to be affected by linkage disequilibrium, therefore, also appearing highly clonal and includes a very limited number of homoplastic sites.

We thank the reviewer for suggesting that we check for homoplasies in our new genomes. We find that none of the five uniquely identified SNPs in RT5 appear to be shared with other branches in the phylogeny. This is now mentioned in our main manuscript as follows (lines 185-186): “As RT5 possesses five unique non-homoplastic SNPs, we checked those for identity to RT6...”

Taken together, our current and previously demonstrated results show that *Y. pestis* is, and has been, a highly clonal, monomorphic, pathogen, whose genetic history can be largely discerned through whole genome phylogenetic analysis.

Politomies: The authors discuss extensively a polytomy dated to around 3,000bp. Polytomies can indicate rapid demographic expansion, or they can simply be caused by insufficient phylogenetic resolution. This polytomy of three branches, includes one that leads to the Angola (0.PE3) strain as the only tip. As the authors mention in the paper, Angola is a very atypical strain with no known close relative, an excess of private SNPs and an obscure origin, and it might even be a strain that has been passaged in a lab. The authors wrote in the discussion:

“In contrast to 0.PE4, the isolation history and natural provenance of 0.PE3 are not published, and since the only available genome representative of this lineage is atypical with a high number of private SNPs and a unique genome architecture (Fig. 2b), we have excluded it from further interpretations.”

Given the statement above that Angola was excluded from further interpretations, I was surprised to see it included in Figure 2b, and even more so for the authors to discuss at length a polytomy involving the Angola strain (and that would not exist without the inclusion of Angola). If the authors have no confidence in the phylogenetic placement of Angola, I recommend they ignore any mention of this polytomy.

We have followed the reviewer's suggestion and omitted the Angola (0.PE3) genome from our analyses (see Figure 2c), as well as any reference to this isolate in our main text. However, after construction of a new phylogenetic tree we observe that our described polytomy/radiation exists even after the exclusion of the Angola isolate. This polytomy is now composed of three lineages instead of four.

Since we use a complete deletion approach in our phylogenetic reconstruction we investigated whether the detected polytomy was caused by a collapsing of branching points at that position, which may have resulted by the exclusion of SNPs that appear as missing data in one or more genomes within our dataset. We constructed an additional ML tree using all variable positions (including missing data from all isolates), and nonetheless see the presence of a polytomy relating to the node that gave rise to RT5 (now shown in Supplementary fig. 3). We therefore propose that this event likely does represent a rapid expansion of lineages dating around 4,000 years ago.

Regardless, we agree that demonstrating an epidemic event during that time would perhaps require more data, and therefore we have followed the reviewer's suggestion and omitted from our discussions any suggestion of a direct relation to an epidemic. Our altered text states as follows: *"Given that previous research has proposed a relationship between rapid Y. pestis expansions and historical plague epidemics in humans², future investigations of lineage diversity from modern and ancient sources may reveal additional details on this ancient radiation event."*

Minor Comments

Figure 2b: What do stars denote in figure 2b. The star symbol is sometimes used to indicate statistical support higher than 95%, but that is unlikely to be the case here, as bootstrap support values up to 99% are indicated in full on some nodes.

In our previous version an asterisk (*) was made to denote a 100 bootstrap value. In our new version it is now changed to denote bootstrap values of 95 or higher (see Figure 2b, and Supplementary fig. 3).

Write-up: The paper needs to make it clearer what are the factual results, the inferred results and the speculations.

In an attempt to accommodate the suggestions by Reviewer #2 we have implemented changes in our discussion to specify the parts in which we describe factual results, inferred results and where we make suggestions for future work. Here is an outline of all paragraphs in our discussion, as well as the points and changes made within them:

In Paragraph 1 (starting at line 306) we discuss our new human DNA findings relatively to previously published work. We believe all statements made in this section derive from well-demonstrated results within our study or in previously published studies. No changes have been made to this section.

In Paragraph 2 (starting at line 320) we discuss our *Y. pestis* genomic results, as well as the inferred divergence dates of RT5 as part of a polytomy of three lineages. Relating to a previous comment by Reviewer #2 we have omitted a link between this polytomy and a

possible epidemic, and have included a concluding remark to this section as follows: *“Given that previous research has proposed a relationship between rapid *Y. pestis* expansions and historical plague epidemics in humans², future investigations of lineage diversity from modern and ancient sources may reveal additional details on this ancient radiation event.”*

Paragraph 3 (starting at line 338) includes a description of all three lineages that derive from the polytomy first identified in our study. All statements describing the disease potential of two previously published lineages have been demonstrated by either a direct isolation of the bacterium from disease cases, sylvatic rodent reservoirs, or in association to historical epidemics where the isolated strains derive from well established historical contexts. The characterisation of our RT5 lineage and its disease potential is made solely based on the genomic results, which we believe is clearly stated within our text. One such example within this paragraph is the following: *“Despite the variation in pathogenicity of 0.PE4 isolates, the phylogenetically related RT5 strain seems unaffected by genetic alterations/disruptions associated with reduced virulence (Fig. 3, Supplementary fig. 9 and Supplementary Table 9) and therefore, we have no evidence that this strain had a decreased virulence in humans.”*

Paragraph 4 (starting at line 360) includes a synthesis of our genomic, phylogenetic and dating results in relation to published evidence from modern lineages 0.PE2 and 0.PE7, which are phylogenetically ancestral to RT5. We utilize their previously published isolation histories and our new molecular dating results to hypothesize that flea-adapted *Y. pestis* strains likely existed before 5,000 years ago, since the divergence date of the bubonic-plague-associated 0.PE7 is >5,000 BP. Given that lineage 0.PE7 seems to be relatively rare today and, to-date, no ancient isolate has been associated with this lineage, we have modified the concluding remark to this paragraph which now states as follows: *“It is, therefore, tempting to hypothesize that efficient flea-adaptation in *Y. pestis*, as well as the potential for bubonic disease, might have evolved earlier than 5,000 years ago.”*

Paragraph 5 (starting at line 391) is our concluding paragraph, in which we provide the potential impact of our study in light of recent findings on human migrations during the Bronze Age. We believe we clearly state the basis of our hypotheses and emphasize that more evidence from future studies would be required to discern our suggested scenarios. For example we mention: *“Whether these lineages had equal prevalence among human populations, and the extent to which human practices contributed to their dissemination, are concepts requiring further investigation”,* as well as *“Additional Bronze Age/Iron Age genomes could provide further insights into the early stages of *Y. pestis* evolution, and help pinpoint key events that contributed to the high virulence and spread of one of humankind’s most notorious pathogens.”* Therefore, the information within this paragraph has been kept as is.

Reviewer #3 (Remarks to the Author):

The oldest evidence for *Yersinia pestis* was found in several Late Neolithic-Bronze Age individuals across Eurasia in a previous work by Rasmussen et al. (2015), a finding that was suggestive of a previously unknown epidemic coming into Europe along with the steppe pastoralists at the end of the Neolithic. However, the lack of virulence associated plasmids in these ancient strains suggested that the Bronze Age epidemic could not trigger the most deadly form of the disease, the so-called “bubonic” form that is able to be transmitted to humans through bites from infected fleas. Therefore, it was yet not well understood when *Yersinia pestis* acquired its modern, high pathogenicity.

In this work the authors have found evidence of the pathogen in nine Late Bronze Age individuals from the Central steppe region (Samara, in Russia) and retrieved the complete genome of *Y. pestis* in two individuals by combining two different methods (plus the complete human genome of one of them). The analysis of the pathogen showed it already had the virulence factors and therefore, showed a more-derived position in the *Yersinia*

phylogenetic tree than other contemporaneous samples. The analysis of the human genome (and also the Y chromosome) placed this individual along the steppe nomads that transformed the genetic European landscape by the end of the Neolithic. They also dated by coalescent methods the origin of the pathogenic lineage and found a date of at least 4,000 years ago, 1,000 years older than previously assumed because of its absence in previous Bronze Age samples. In conclusion, this work proves that both *Yersinia* lineages existed contemporaneously along the Central Asian steppes.

I found this an interesting paper that solves the obvious question of the date of the pathogenic strain and that therefore can be of interest to researchers in different fields, from experts on infectious diseases to evolutionary biologists and archaeologists. The methodology used, both experimentally and computationally, are the appropriate ones. In addition, the paper is clearly written.

I don't have major concerns or criticisms. Maybe the authors could mention this paper: <https://www.biorxiv.org/content/early/2017/05/09/135962> in which there is a general analysis of hundreds of Late Neolithic and Bronze Age samples across Western Europe and here they found evidence of a 90 population replacement of the British Neolithic populations with the arrival, from The Netherlands (or from surrounding regions) of the Bell Beakers, something that is clearly a following up of the east-west expansion that brought the steppe ancestry into Eastern and Central Europe. In the light of their findings, it is likely that a pathogenic *Yersinia pestis* could contribute to the extreme social and demographic turn-over detected in this work.

We thank Carles Lalueza-Fox for his encouraging review and constructive comments. We have mentioned the pre-print by Olalde, *et al.* in the introduction of our main manuscript, and it now appears as reference number 25.

Also, the authors mention that the *Yersinia* reads harbour the signs of post-mortem damage at the ends of the reads that are proxy of authenticity. These damage patterns correlate with the age of the sample (Sawyer *et al.* 2012) and the authors have now generated *Yersinia* reads from quite different periods, along the last 4,000 years. Could it be possible to create a Figure to see if the extent of damage pattern is correlated with temporal dates on the pathogen strains? I have seen some people expressing doubts on the paleopathogen studies, so an evidence of this line would be nice, although I am not sure if there is enough data.

We agree that such a metric for pathogen DNA authentication would be highly valuable and have attempted to perform the proposed analysis (see figure below) for the non-UDG treated ancient *Y. pestis* data available to-date. However, we unfortunately do not currently trust that the amount of material available and the temporal depth in our sample makes this analysis meaningful. Sawyer *et al.*¹⁰ show that a minimum of 10% terminal cytosine deamination should be present in samples equal to or older than 500 years. Indeed we see >10% damage in all material analysed to date, even for those younger than 500 years, however we observe a big gap in the available data between 1,000 and 3,000 years BP, a fact that complicates the interpretation and power of the observed result. In addition, we see high variation in the amount of damage within individual time periods, and, therefore, an extrapolation of a specimen's age from the amount of deamination is likely not feasible.

Since we only included the UDG-half damage plots in the manuscript, we agree that also including the non-UDG plots would be a meaningful line of evidence for the authenticity of our genome, as well as a useful metric for future studies. Therefore, we have now included the non-UDG damage patterns for individual RT5 in Supplemental fig. 1, for which we find 27% of terminal deamination.

References

- 1 Rasmussen, S. *et al.* Early Divergent Strains of *Yersinia pestis* in Eurasia 5,000 Years Ago. *Cell* **163**, 571-582, doi:10.1016/j.cell.2015.10.009 (2015).
- 2 Cui, Y. *et al.* Historical variations in mutation rate in an epidemic pathogen, *Yersinia pestis*. *Proceedings of the National Academy of Sciences of the United States of America* **110**, 577-582, doi:10.1073/pnas.1205750110 (2013).
- 3 Rambaut, A., Lam, T. T., Max Carvalho, L. & Pybus, O. G. Exploring the temporal structure of heterochronous sequences using TempEst (formerly Path-O-Gen). *Virus evolution* **2**, vew007, doi:10.1093/ve/vew007 (2016).
- 4 Kumar, S., Stecher, G. & Tamura, K. MEGA7: Molecular Evolutionary Genetics Analysis Version 7.0 for Bigger Datasets. *Molecular biology and evolution* **33**, 1870-1874, doi:10.1093/molbev/msw054 (2016).
- 5 Baele, G., Lemey, P. & Vansteelandt, S. Make the most of your samples: Bayes factor estimators for high-dimensional models of sequence evolution. *BMC bioinformatics* **14**, 85, doi:10.1186/1471-2105-14-85 (2013).
- 6 Drummond, A. J. & Rambaut, A. BEAST: Bayesian evolutionary analysis by sampling trees. *BMC evolutionary biology* **7**, 214 (2007).
- 7 Stadler, T., Kuhnert, D., Bonhoeffer, S. & Drummond, A. J. Birth-death skyline plot reveals temporal changes of epidemic spread in HIV and hepatitis C virus (HCV). *Proceedings of the National Academy of Sciences of the United States of America* **110**, 228-233, doi:10.1073/pnas.1207965110 (2013).

- 8 Bouckaert, R. *et al.* BEAST 2: a software platform for Bayesian evolutionary analysis. *PLoS computational biology* **10**, e1003537, doi:10.1371/journal.pcbi.1003537 (2014).
- 9 Andrades Valtuena, A., *et al.* The Stone Age plague and its persistence in Eurasia. *Current Biology* **27**, 1-9, doi:10.1016/j.cub.2017.10.025 (2017).
- 10 Sawyer, S., Krause, J., Guschanski, K., Savolainen, V. & Paabo, S. Temporal patterns of nucleotide misincorporations and DNA fragmentation in ancient DNA. *PloS one* **7**, e34131, doi:10.1371/journal.pone.0034131 (2012).

Reviewers' comments:

Reviewer #2 (Remarks to the Author):

I found the revision improved and the authors have satisfactorily addressed my previous concerns, with one exception (see below).

The authors decided not to test for recombination citing evidence from the literature for the absence thereof, which is more ambiguous than the authors might believe, The Cui et al. 2013 paper based on modern isolates and two medieval ones did in fact identify about 5% of homoplastic sites, which were removed from their phylogenetic reconstructions.

The approach in Rasmussen et al. 2015 to test for recombination is not adequate. A decay of linkage disequilibrium (LD) across physical distance is a hallmark of recombination in eukaryotes.

Recombination in bacteria is more akin to gene conversion and does not lead to a decay in LD. The correct way to test for recombination in primarily clonal bacteria is to test for the presence of homoplasies.

It remains unclear when *Y. pestis* became "clonal" and residual recombination in the basal branches could dramatically affect age estimates. I suggest the authors test for homoplasies in their dataset, which is straightforward to do, and in case they found some, they should rerun the phylogenetic analyses without homoplastic sites.

Reviewer #3 (Remarks to the Author):

The authors have answered all my comments on the previous manuscript.

Response to Reviewers

Reviewer #2 (Remarks to the Author):

I found the revision improved and the authors have satisfactorily addressed my previous concerns, with one exception (see below).

The authors decided not to test for recombination citing evidence from the literature for the absence thereof, which is more ambiguous than the authors might believe, The Cui et al. 2013 paper based on modern isolates and two medieval ones did in fact identify about 5% of homoplastic sites, which were removed from their phylogenetic reconstructions.

The approach in Rasmussen et al. 2015 to test for recombination is not adequate. A decay of linkage disequilibrium (LD) across physical distance is a hallmark of recombination in eukaryotes. Recombination in bacteria is more akin to gene conversion and does not lead to a decay in LD. The correct way to test for recombination in primarily clonal bacteria is to test for the presence of homoplasies.

It remains unclear when *Y. pestis* became "clonal" and residual recombination in the basal branches could dramatically affect age estimates. I suggest the authors test for homoplasies in their dataset, which is straightforward to do, and in case they found some, they should rerun the phylogenetic analyses without homoplastic sites.

We thank the reviewer for his/her comments, as we believe that our manuscript has improved as a result of their suggestions.

We recognise that our "Methods" section lacked information on the regions that we exclude from our comparative SNP analysis, which in fact are all previously defined non-core regions, mRNAs, tRNAs, tmRNAs and homoplasies^{1,2} (see lines 504 - 508 in main manuscript). In addition, we updated the number of variant sites in our dataset, as the original number did not account for the removal of low coverage genomes from our phylogenetic inference. When accounting for these changes, our dataset of 179 genomes (159 modern and 20 ancient) comprises of a total 3,849 SNP positions.

We have followed the reviewer's suggestions and tested for evidence of recombination and homoplasies in our entire dataset. For this we used the program ClonalFrameML³, where a full genome alignment of all isolates used in this study as well as a maximum likelihood phylogeny generated using RaxML⁴ were used as input. We performed the standard model analysis using the Baum-Welch Expectation-Maximisation algorithm (-em option) without considering missing data (-ignore_incomplete_sites option), as we also exclude missing sites for our phylogeny and molecular dating. ClonalFrameML identified 28 additional variant sites (see the table and output figure displayed below) that could potentially be problematic to our phylogenetic inference and divergence date estimates. 26 of these sites seem to derive from potentially recombining regions as defined by ClonalFrameML and two appear homoplastic (table below also appears as Supplementary table 11 in our manuscript). After excluding these sites our dataset comprised of 3,821 total SNP positions. The analysis described here now appears in our "Methods" section (lines 508 - 514).

In addition, we repeated the phylogenetic (see Figure 2 and Supplementary figures 3 & 4 in our manuscript), and molecular dating analysis (see Supplementary figure 5, 6 & 7 and Supplementary table 9) using the new SNP dataset, which yielded an almost identical result to the one reported in the previous versions of our manuscript. More specifically, our estimates suggest a divergence date of 5,727y BP (HPD95%: 4,909-6,842) as opposed to 5,730y BP (HPD95%: 4,906-6,874) reported in the previous version of our manuscript.

Table - (appears as Supplementary table 11 in main manuscript). Newly identified SNPs that were excluded from comparative variant calling analysis in addition to previously defined regions.

Position in CO92	Reference	Variant	Description	Strains identified in
1,029,500	A	G	SNP*	Georgia 1413 (Zhghenti et al. , 2015)
1,029,502	A	C	SNP*	Georgia 1413 (Zhghenti et al. , 2015)
1,029,503	A	C	SNP*	Georgia 1413 (Zhghenti et al. , 2015)
1,361,705	T	C	SNP*	6304 (Kislichkina et al. ,2015)
1,361,707	A	G	SNP*	6304 (Kislichkina et al. ,2015)
1,361,719	G	T	SNP*	6304 (Kislichkina et al. ,2015)
1,687,299	G	T	SNP*	6904 (Kislichkina et al. ,2015)
1,687,300	T	C	SNP*	6904 (Kislichkina et al. ,2015)
1,687,301	T	C	SNP*	6904 (Kislichkina et al. ,2015)
3,489,416	G	A	SNP*	1.ANT1_UG05-0454
3,489,419	G	A	SNP*	1.ANT1_UG05-0454
3,489,428	G	T	SNP*	1.ANT1_UG05-0454
3,489,429	A	T	SNP*	1.ANT1_UG05-0454
3,860,629	A	G	SNP*	0.PE7b_620024
3,860,637	A	C	SNP*	0.PE7b_620024
3,860,639	A	T	SNP*	0.PE7b_620024
4,273,931	C	A	SNP*	0.ANT3a_CMCC38001
4,273,933	C	T	SNP*	0.ANT3a_CMCC38001
4,273,941	A	T	SNP*	0.ANT3a_CMCC38001
4,273,942	A	T	SNP*	0.ANT3a_CMCC38001
4,355,693	C	A	SNP*	6216 (Kislichkina et al. ,2015)
4,355,759	T	C	SNP*	6216 (Kislichkina et al. ,2015)
4,355,760	A	G	SNP*	6216 (Kislichkina et al. ,2015)
3,939,869	T	A	SNP*	6304 (Kislichkina et al. ,2015)
3,939,870	T	C	SNP*	6304 (Kislichkina et al. ,2015)
3,939,872	T	C	SNP*	6304 (Kislichkina et al. ,2015)
358,876	A	C	homoplastic SNP	shared between LNBA, 0.PE4 and 2.MED3i
1,805,037	C	A	homoplastic SNP	shared between 0.PE2 and 0.PE4

*SNPs that appear within potentially recombining regions as defined by ClonalFrameML.

Figure - ClonalFrameML output illustration depicting a maximum likelihood tree with adjusted branch lengths and a heat map representation of potential regions that may be affected by recombination. Previously defined non-core and low complexity regions, tRNAs, mRNAs and tmRNAs^{1,2} appear in grey (regions displayed are ≥ 1000 bp). Light blue colour represents reference calls, whereas dark blue regions represent potential recombination arrears defined by ClonalFrameML. In addition, non-homoplasic SNPs appear white, and homoplasic sites appear from a range of yellow (least) to red (most) based on their degree of homoplasy.

References

- 1 Morelli, G. *et al.* Yersinia pestis genome sequencing identifies patterns of global phylogenetic diversity. *Nature genetics* **42**, 1140-1143, doi:10.1038/ng.705 (2010).
- 2 Cui, Y. *et al.* Historical variations in mutation rate in an epidemic pathogen, Yersinia pestis. *Proceedings of the National Academy of Sciences of the United States of America* **110**, 577-582, doi:10.1073/pnas.1205750110 (2013).
- 3 Didelot, X. & Wilson, D. J. ClonalFrameML: efficient inference of recombination in whole bacterial genomes. *PLoS computational biology* **11**, e1004041 (2015).
- 4 Stamatakis, A. RAxML version 8: a tool for phylogenetic analysis and post-analysis of large phylogenies. *Bioinformatics* **30**, 1312-1313, doi:10.1093/bioinformatics/btu033 (2014).

REVIEWERS' COMMENTS:

Reviewer #2 (Remarks to the Author):

The authors have addressed all my concerns. As such, I recommend acceptance of the paper.